# Investigating the Flow Characteristics of Superhydrophobic U-Shaped Microchannels

Zhi Tao [1], Weidong Fang [1], Haiwang Li [1], Tiantong Xu [1,*], Yi Huang [1], Hanxiao Wu [2] and Murun Li [1]

1 National Key Laboratory of Science and Technology on Aero-Engine Aero-Thermodynamics, Research Institute of Aero-Engine, Beihang University, Beijing 100191, China
2 Beijing Institute of Astronautical Systems Engineering, China Academy of Launch Vehicle Technology, Beijing 100076, China
* Correspondence: xutiantong@buaa.edu.cn

**Abstract:** Hydrophobicity has been widely reported for its superior behavior in drag reduction, self-cleaning, and anti-corrosion in many areas. Especially in engineering design, the research of the unique property of the slip flow with complex flow patterns is essential for practical applications. In this paper, the flow characteristics of a superhydrophobic U-shaped microchannel are systematically investigated, as the curved part is a fundamental component in microfluids. A slip model is established based on theoretical and experimental solutions. Various types of U-shaped microchannels, radii of curvature, and contact angles are studied with a wide range of Reynolds numbers from 0 to 300. We propose a velocity distribution to examine the non-uniformity of slip velocity on the cross-section. This imbalance is improved with an increase in the apparent contact angle and flow rate, and a decrease in the radius of curvature. The secondary flow and vortices generated by the centrifugal force are enhanced, and their positions are changed due to the slippery boundary. The results show a considerable drag reduction from 10% to 40% with different contact angles. The variation of curvature does not have a decisive impact on the final performance when the surface wettability maintains a steady state. Our research elucidates the physical principle of the slip model in curved channels, showing extensive applications of hydrophobicity in the design of complex microchips and the optimization strategy of heat transfer systems.

**Keywords:** microchannel; U-shaped; superhydrophobic; drag reduction

## 1. Introduction

With the rapid development of the semiconductor industry, aeronautics, and astronautics, more stringent requirements have been proposed for microscale cooling and heat transfer technology. Statistically, the heat flux density of microchips has rapidly increased since the 1980s [1]. Microchannel heat sinks have become the key to the development of refrigeration technology due to their several advantages, such as high efficiency and small dimensions. For example, the distribution of micro-ribs [2,3], structural parameters for cavities [4,5], and effects of longitudinal vortices [6–8] in microchannels have recently become popular topics in microfluidics. As well as improving the heat transfer efficiency, microstructures also increase the friction resistance.

In recent years, inspired by the unique water-repellent properties of the lotus leaf [9], many scholars have begun to apply hydrophobicity to the fabrication of microchannels [10–12]. Initially, liquid slippage on a hydrophobic surface was speculated to be caused by an intermediate gas layer between the liquid–solid interface rather than the actual sliding of liquid molecules along the solid surface [13]. Another study showed that the boundary conditions on the hydrophobic interface are not constant but variable due to the alternating arrangement of the solid–liquid and liquid–gas interfaces [14,15]. The effective slip length was proposed and estimated, which became the connection between liquid slippage and drag reduction.

The relationship between the channel pressure drop and the slip length was obtained by a numerical method for a fully developed laminar flow on a flat plate [16,17]. Compared with the classical microtube, a superhydrophobic microtube with hydraulic diameters from 0.45 to 0.87 mm can reduce the pressure loss by 60% [18]. However, only straight microchannels with superhydrophobic surfaces have been previously considered. The slip flow with complex flow patterns should also be explored in detail in engineering design. The U-shaped microchannel is an important component in designing microfluidic devices [19]. Understanding the flow property in a U-shaped channel on the microscale has been attempted by many scholars. A considerable number of achievements have been made for no-slip U-shaped channels. For example, the flow characteristics and velocity distribution have been found to be determined in microtubes with different cross-sectional shapes and roughness [20]. The accelerated corrosion effect in curved channels of the flow was also investigated through numerical simulation, and the degree of severe wear at different positions was analyzed [21]. In summary, both the superhydrophobic property and U-shaped channel play an important role in microfluidics. However, their comprehensive performance has not been systematically evaluated, which may contribute to the bioanalysis [22], microdevices [23], and heat sinks [24].

In this study, the flow characteristics of a super-hydrophobic U-shaped microchannel are investigated. In particular, with an increase in the Reynolds number and apparent contact angle, the three-dimensional non-uniformity of slip velocity and the secondary flow effect are analyzed under different radii of curvature. To compare the drag reduction behavior in U-shaped microchannels, a dimensionless factor representing the difference in pressure loss is introduced. The potential contribution for further application in micro heat sinks and microfluidic chips is also discussed.

## 2. Superhydrophobic Theory

Superhydrophobic theory consists of two parts: hydrophobic surface wettability and interfacial hydrodynamic slippage, which represent its static and dynamic properties, respectively.

### 2.1. Hydrophobic Surface Wettability

Hydrophobicity is often measured by surface wettability, which is mainly evaluated by the contact angle between the liquid and the solid surface. In 1805, Young proposed an interfacial chemical equation that describes the contact angle on a smooth plane:

$$cos(\theta_e) = (\gamma_{sg} - \gamma_{sl})\gamma_{lg} \tag{1}$$

where $\gamma_{sg}$, $\gamma_{lg}$, and $\gamma_{sl}$ are the surface tension of the solid–gas, liquid–gas, and solid–liquid surface, respectively, and $\theta_e$ is the intrinsic contact angle of the material surface. However, the intrinsic contact angle is sometimes different from the observed contact angle, especially for rough surfaces with microstructures. Wenzel et al. [25] and Cassie et al. [26] proposed a physical model for the relationship between the apparent contact angle and the intrinsic contact angle, respectively.

The Wenzel model assumes that the grooves formed by microstructures are always filled with liquid on the hydrophobic surface. The actual solid–liquid contact area is larger than a smooth surface, resulting in the hydrophilicity or hydrophobicity of rough surfaces, as shown in Figure 1a.

Then, the apparent contact angle is calculated as:

$$cos\theta_c = cos\theta_e \cdot r \tag{2}$$

where $\theta_c$ is the apparent contact angle, and $r$ is the ratio of the actual contact area to the apparent contact area. In the Wenzel model, if the smooth surface is hydrophobic, microstructures will enhance its performance.

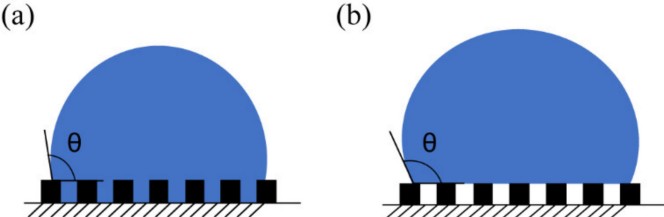

**Figure 1.** Hydrophobic surface models: (**a**) Wenzel model and (**b**) Cassie model.

The Cassie model presumes that there is still some gas trapped in the grooves that prevents the liquid from wetting the cavities, as shown in Figure 1b. The contact surface is a three-phase composite interface composed of solid–liquid and gas–liquid interfaces. In this model, the apparent contact angle for a composite interface is given by:

$$cos\theta_c = (1 - f_a)(1 + cos\theta_e) - 1 \tag{3}$$

Here, $f_a$, which is the proportion of gas–solid contact surface area, is the most important factor in controlling the apparent contact angle. A higher value of $f_a$ contributes to a larger apparent contact angle and more significant hydrophobicity.

The selection of these models must be considered according to the material properties, microstructures, and other components. In practical applications, the gas exchange, random vibration, heating, and other factors [27] may affect the stability of the composite interface in the Cassie model and transform it into the Wenzel model.

### 2.2. Interfacial Hydrodynamic Slippage

Hydrophobicity not only contributes to the surface wettability of liquid droplets but also affects the flow characteristics of the three-phase composite interface, which is mainly represented by the drag reduction in the microchannel. To evaluate the drag reduction, the slip length is typically observed and analyzed. The slip length is defined as the ratio of slip velocity to the wall shear rate, or the virtual distance obtained by linear extrapolation of the velocity at the boundary, as shown in Figure 2.

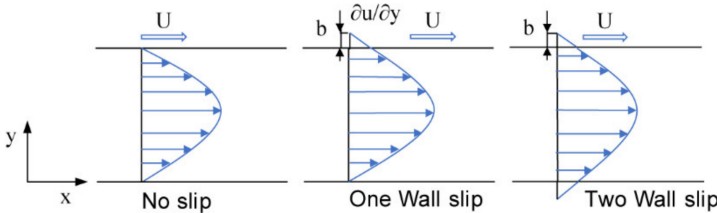

**Figure 2.** Slip length on an immersed slippery boundary.

The slip length, which determines the performance of a hydrophobic microchannel, is an important parameter for drag reduction in superhydrophobic microchannels [28]. Therefore, the slip length has been identified and analyzed by several scholars. Performing numerical simulation with the lattice Boltzmann method, the slip length is found to be only related to the microstructures on the hydrophobic surface and liquid properties, which is independent of the experimental pressure gradient [29]. However, the drag reduction of flowing through a tube with hydrophobic surfaces has been experimentally investigated [11]. The results show that the drag reduction varies with the development of the Reynolds number, because the friction resistance generated by pinning on the liquid–solid interface affects the slippery boundary, which means that the slip length varies with the variation of flow conditions. Furthermore, another study indicated that the slip length of a superhydrophobic microchannel increases as the hydraulic diameter increases, which is attributed to the influence of surface curvature, microstructure, and experimental pressure [18].

The contradiction of the above conclusions is mainly caused by the variation in flow conditions and instability of the local gas–liquid interface in the actual experimental process, which may transform the surface wettability from the Cassie model to the Wenzel model. This phenomenon finally leads to pressure drop fluctuation and instability of the slip length. In fact, in a steady laminar flow, the slip length is mainly related to the microstructures and the chemical composition of the surface as the surface wettability still remains in the Cassie state. Therefore, a uniform superhydrophobic surface is fabricated in a microchannel to investigate the relationship between slip length and the proportion of gas–solid contact surface area [30]. Figure 3 shows the relationship between apparent contact angle, slip length, and proportion of gas–solid contact surface area.

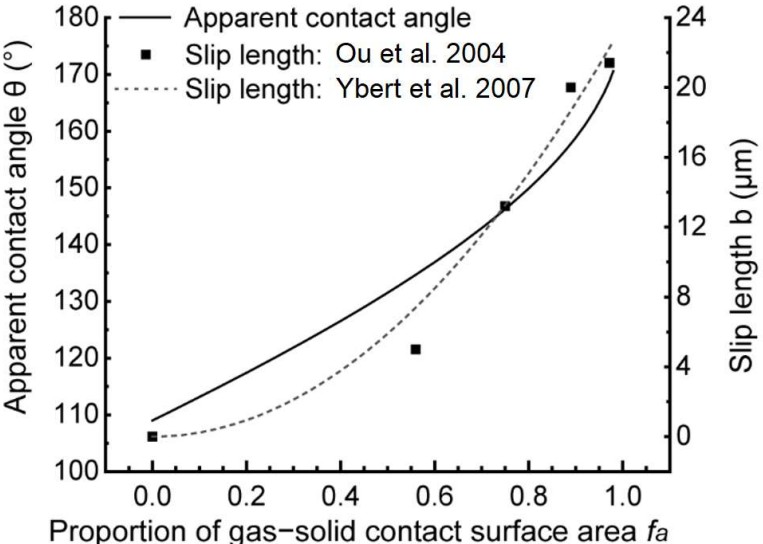

**Figure 3.** Relationship between apparent contact angle, slip length, and proportion of gas–solid contact surface area [30,31].

Despite some differences between the results reported in the literature [31], the variation range and trend of the slip length are essentially the same, which is an important reference for slippery boundary modification in the numerical model. To note, other external disturbances also have an impact in the hydrophobic microchannel, leading to an unpredictable change of the slip length. The disappearance of slip length and decline of hydrophobic properties may occur when the external disturbance cannot be neglected, which will be another independent work on transition of the hydrophobic surface. In general, the flow in U-shaped microchannels usually presents a steady laminar flow state in this model, indicating that the hydrophobic surface remains in the Cassie state. Therefore, in this model, we assume that the slip length remains constant in working conditions.

### 3. Numerical Model

In this study, a finite element method model in COMSOL of a superhydrophobic U-shaped microchannel was established, and its geometry is shown in Figure 4. In this model, water, as a common fluid, was adopted, of which the mean free path, $\lambda$, is $5.752 \times 10^{-8}$ m. The Knudsen Number can be calculated as follows:

$$K_n = \frac{\lambda}{d} = \frac{5.752 \times 10^{-8}}{3 \times 10^{-4}} = 1.9173 \times 10^{-4} < 0.001 \tag{4}$$

Therefore, the laminar flow model was adopted, and the working fluid was regarded as continuous, steady, and incompressible. The reference temperature was set to 293.15 K. Since it is a single-phase flow on the microscale, gravity and surface force in the channel can be ignored in the calculation. Equation (5) is the simultaneous equations, which consists

of a continuity equation and a vector equation, representing conservation of mass and momentum:

$$\begin{cases} \rho\left(\vec{u}\cdot\nabla\right)\vec{u} = \nabla\cdot\left[-p\vec{I} + \vec{K}\right] + \vec{F} \\ \rho\nabla\cdot\vec{u} = 0 \end{cases} \tag{5}$$

where $\rho$ is the density, $\vec{u}$ is the velocity vector, $p$ is the pressure, and $\vec{I}$ is the space tensor. $\vec{K}$ and $\vec{F}$ are the viscous stress tensor and the volume force vector, respectively.

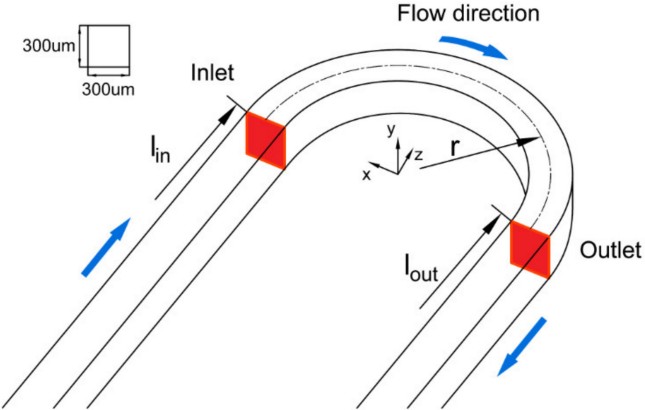

**Figure 4.** Geometry of the U-shaped microchannel.

The dimensions of the U-shaped microchannel were a width, $a$, of 300 μm and a depth, $b$, of 300 μm, and the radius of curvature, $r$, was varied as 0.5, 1.0, and 1.5 mm, respectively. Slip boundary conditions were applied around the wall. Both the inlet length, $l_{in}$, and the outlet length, $l_{out}$, were properly extended to ensure that the inlet flow was fully developed, avoiding conflict between the constant-pressure outlet boundary and the non-uniform flow.

Four contact angles: 135°, 143°, 150°, and 165°, were selected according to the current superhydrophobic surface fabrication methods, as described in Table 1.

**Table 1.** Apparent contact angle in different superhydrophobic surface fabrication methods.

| Superhydrophobic Surface Fabrication Method | Apparent Contact Angle |
|---|---|
| Novel polydimethylsiloxane (PDMS)-based microchannel [32] | 135° |
| $C_{18}H_{37}Cl_3Si$ | 143° |
| $C_{18}H_{37}Cl_3Si$ and hierarchical structured surfaces [33] | 150° |
| Vertically aligned carbon nanotube forests modified with conformal silicone coating [34] | 165° |

According to the relationship illustrated in Figure 3, the slip lengths of the apparent contact angles were estimated to be 5.3, 11.1, 15.7, and 21.0 μm, respectively. Due to the constant slip length under different flow conditions in the steady flow, the slippery boundary was modified as:

$$\vec{u}_s\cdot\vec{n} = 0 \tag{6}$$

$$\vec{\tau}_{wall} = -\frac{\mu}{b}\vec{u}_s \tag{7}$$

where $\vec{u}_s$ and $b$ are the slip velocity and slip length, respectively, $\mu$ is the dynamic viscosity of the fluid, and $\vec{\tau}_{wall}$ is the wall shear stress in the superhydrophobic channel.

## 4. Simulation Validation

The accuracy and validity of the slippery boundary determines the results. For the no-slip condition, numerical results were compared with theoretical solutions with

different aspect ratios in the rectangular channels. Next, a precise numerical model for a superhydrophobic straight channel from the existing literature was established. The experimental data and simulation results were evaluated to confirm the effectiveness of the slippery boundary.

### 4.1. Theoretical Solution

To perform a comprehensive comparison and analysis, we designed a new data processing method to evaluate the drag reduction performance for the U-shaped microchannels. Therefore, it is necessary to describe the flow characteristics of superhydrophobic U-shaped channels, which are expressed by Poiseuille number, *Po*, and drag reduction rate, *p*:

$$d = 4ab/2(a+b) \tag{8}$$

$$l = \pi r \tag{9}$$

$$Re = \frac{\rho \overline{U} d}{\mu} \tag{10}$$

where $d$ is the hydraulic diameter, $l$ is the centerline length, which is equal to the semicircle perimeter with radius of curvature $r$, and $\overline{U}$ denotes the average speed at which the fluid moves in the microchannel.

$$\Delta P = P_{in} - P_{out} \tag{11}$$

$$Po = fRe = \frac{\Delta P}{0.5\rho \overline{U}^2} \frac{d}{l} Re = \frac{P_{in} - P_{out}}{0.5\rho \overline{U}^2} \frac{d}{\pi r} Re \tag{12}$$

$P_{in}$ and $P_{out}$ are the inlet and outlet pressure of the channel, respectively.

$$p = \frac{Po_{no-slip} - Po_{slip}}{Po_{no-slip}} \times 100\% \tag{13}$$

Here, $Po_{no-slip}$ and $Po_{slip}$ are the *Po* of no-slip and superhydrophobic channels, respectively.

Numerical analysis of a fully developed laminar flow in straight rectangular channels has shown that the velocity component was zero in the cross-section [35]. Therefore, the simplified momentum equation of the flow direction ($z$ direction) can be expressed as:

$$\mu \left( \frac{\partial^2 w}{\partial x^2} + \frac{\partial^2 w}{\partial y^2} \right) - \frac{dp}{dz} = 0 \tag{14}$$

The laminar flow in a rectangular channel has been extensively studied [36,37]. After dimensionless transformation and integration of the infinite series, the variation of *Po* in the rectangular channel with different aspect ratios, $\gamma(\gamma \geq 1)$, can be obtained according to Equation (15), as shown in Figure 5.

$$\begin{cases} \lambda_n = \frac{(2n+1)\pi}{2} & (n = 0,1,2,3\ldots\ldots) \\ C = -2\sum_{n=0}^{\infty} \frac{(-1)^n}{\lambda_n^3 \cosh(\lambda_n \gamma)} \int_0^1 \cos(\lambda_n x) dx \int_0^\gamma \cosh(\lambda_n y) dy \\ Po(\gamma) = \frac{96(\gamma/(1+\gamma))^2}{1+3C/\gamma} \end{cases} \tag{15}$$

For fully developed laminar flow in a rectangular channel, $Po = 56.9$ and $Po \rightarrow 96$ for $\gamma = 1$ and $\gamma \rightarrow \infty$, respectively. In this study, rectangular channels with aspect ratios of $\gamma = 1.9$ and $\gamma = 4$ were calculated to obtain $Po = 60.78$ and $72.93$, respectively. These results are highly consistent with the theoretical solution, which proves the accuracy of the numerical model, as shown in Table 2. We will adjust the mesh density to adjust the accuracy of the simulation.

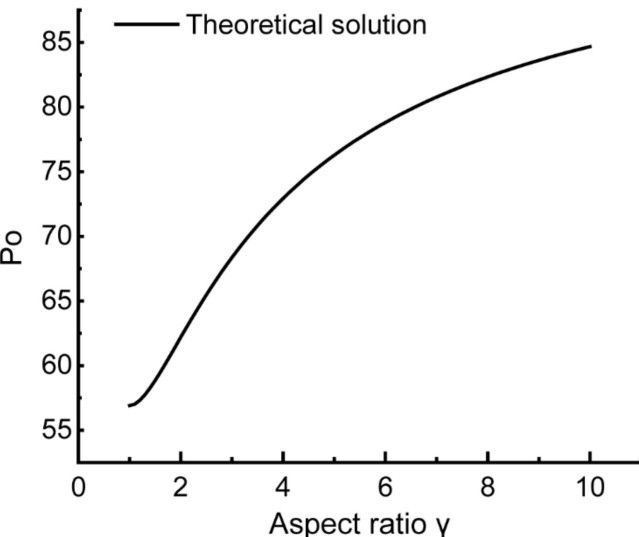

**Figure 5.** Variation of *Po* in rectangular channel with aspect ratio $\gamma$.

**Table 2.** Comparison between the physical and numerical results.

| Aspect Ratio | Theoretical Solution | Numerical Result |
|:---:|:---:|:---:|
| 1 | 56.9 | 56.56 |
| 1.9 | 61.51 | 60.78 |
| 4 | 72.93 | 72.93 |

For mesh-independent validation, a U-shaped microchannel with a radius of 2 mm and a contact angle of 150° was carried out at *Re* = 100. The amount of mesh ranged from $4.68 \times 10^5$ to $1.19 \times 10^6$, increasing by 1.2 times, with a total of 6 sets of grids. The results were compared based on the deviation value (*e*) of the Poiseuille number, as shown in Table 3.

$$e(Po) = \frac{|Po_2 - Po_1|}{Po_1} \times 100\% \tag{16}$$

**Table 3.** Mesh-independent validation.

| No. | Number of Grids | *Po* | *e* (*Po*) |
|:---:|:---:|:---:|:---:|
| 1 | 467,712 | 50.54 | 1.56 |
| 2 | 562,944 | 49.75 | 1.27 |
| 3 | 678,144 | 49.12 | 0.92 |
| 4 | 808,704 | 48.67 | 0.72 |
| 5 | 976,128 | 48.32 | 0.58 |
| 6 | 1,185,024 | 48.04 | |

The deviation value of the No. 5 grid was about 0.58%, which can control the amount of mesh and the accuracy of results. Therefore, the size of No. 5 will be considered as a reference for other channels.

*4.2. Experimental Results*

An experimental investigation of a straight superhydrophobic rectangular microchannel was conducted. In accordance with a previous study, the dimensions were a width of 180 μm, depth of 100 μm, and an apparent contact angle of 135° [32], as shown in Figure 6.

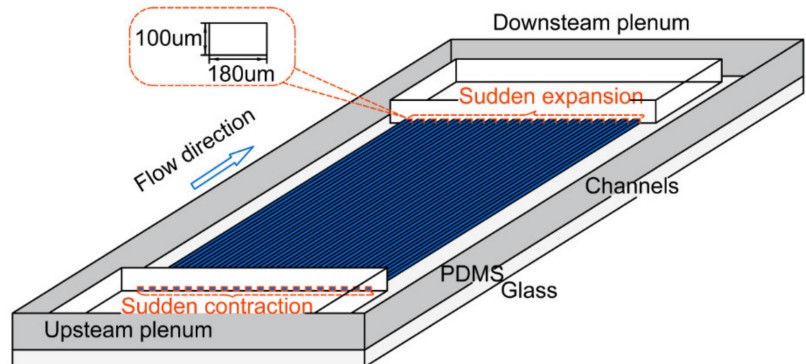

**Figure 6.** Superhydrophobic straight microchannel.

Due to the obvious sudden expansion and contraction regions in this model, the overall pressure drop can be expressed as:

$$\Delta P = \Delta P_{in} + \Delta P_{\text{channel}} + \Delta P_{out} \tag{17}$$

where $\Delta P$ is the overall pressure drop of the channel, $\Delta P_{\text{channel}}$ is the pressure drop generated in the channel, and $\Delta P_{in}$ and $\Delta P_{out}$ are the additional pressure drops at the inlet and outlet of the channel [38], respectively:

$$\begin{aligned}
\Delta P_{in} &= \left[1 - \sigma^2 + K_{in}\right] \times \tfrac{1}{2}G^2 v_f \\
\Delta P_{out} &= -\left[\tfrac{1}{\sigma^2} - 1 + K_{out}\right] \times \tfrac{1}{2}G^2 v_f
\end{aligned} \tag{18}$$

Here, $v_f$ is the fluid-specific volume and $G$ is the mass flux. $\sigma$ represents the ratio of the smaller cross-sectional area to the larger cross-sectional area, and $\dot{m}$ is the mass flow rate. The constant values of $K_{in}$ and $K_{in}$ are 0.44 and 0.77, respectively [39].

The simulation results are in good agreement with the experimental data from the literature. Compared with experimental data, $Po = 59.7$ when $Re = 8.3$, the pressure drop of the numerical model was 591.3 Pa, $Po = 58.6$. The difference between the experiment and the computation was less than 2%, which was probably caused by the slightly different boundary conditions between experimental samples and numerical models. In the experiment, for the convenience of test methods, the design of the array channel was adopted. The effect of a sudden expansion and contraction of the inlet and outlet cannot be neglected regardless of the empirical modified formula. Due to the unpredictable pressure loss of test structures, such as ducting connection, it is reasonable to show some differences between experimental and simulation results.

In general, the numerical results were not only in good agreement with the theoretical solution but were also consistent with previous experiments. Thus, the accuracy of the theoretical model and validity of the slippery boundary have been confirmed in this study.

## 5. Results and Discussion

The results are split into three subsections regarding the slip velocity distribution, secondary flow, and overall drag reduction, which are systematically discussed to evaluate the flow characteristics in a superhydrophobic U-shaped channel.

### 5.1. Slip Velocity Distribution

The slip velocity is considered a typical representative of the hydrophobic property, which can be obtained by derivation from the pressure drop or observed directly by Micron-resolution particle image velocimetry. It is also an important factor affecting the flow characteristics of the microchannel. However, the slip velocity is not only determined by the apparent contact angle but also by other parameters, such as the radius of curvature.

To analyze the slip velocity distribution under different conditions, the middle section of the U-shaped microchannel was selected, as shown in Figure 7.

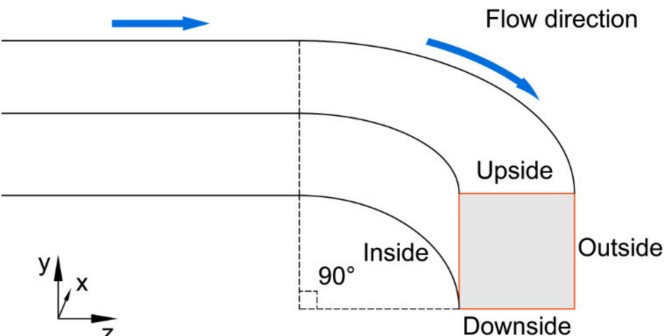

**Figure 7.** U-shaped channel middle cross-section.

The slip velocity distribution varies with Re, radius of curvature, and apparent contact angle, as shown in Figure 8. Compared with the no-slip condition, the centrifugal effect of the flow still has an important influence. The pressure and velocity gradient imbalance were improved at the outside region, which led to the acceleration of slip velocity and the consequent generation of vortices and secondary flows. The corners in the cross-section of the microchannel led to a weaker slip effect, although the slip length remained constant in these corners. In addition, it is clear that the slip velocity in the corner region was much slower than that in the center region, due to the relatively stationary regions of flow induced by the boundary layer effect.

With a decrease in the radius of curvature and an increase in Re, the centrifugal effect of the flow became more significant, which caused the streamline to move outward, and the slip on the wall was enhanced. Furthermore, it can be observed that an increase in the apparent contact angle allowed for a higher slip velocity regardless of the geometric characteristics and flow conditions. The difference between maximum and minimum slip velocity on the wall also increased with the increase of the slip length. There was a difference between the slip velocities on the inside and outside walls as well as the center and corner regions. The imbalance represented by the difference of slip velocity was gradually improved by the development of Re.

The results show that the assumption of uniform slip velocity is not applicable to a U-shaped microchannel, which can be influenced by the geometric characteristics and flow conditions.

*5.2. Secondary Flow*

As opposed to a straight channel, the Dean vortex, which is superposed on the primary flow, is one of the most important secondary flow patterns in a U-shaped channel [40]. The fluid moved forward to the outside of the channel by the centrifugal force in the center and moved to the inside near the wall, which caused the maximum velocity region to shift outward, as shown in Figure 8. To quantify the impact of the superhydrophobic surface, a factor representing the intensity of the secondary flow was introduced as:

$$S = U_{\parallel}/\overline{U} \tag{19}$$

where $\overline{U}$ is the mean velocity magnitude of the microchannel, and $U_{\parallel}$ is the velocity magnitude parallel to the cross-section. The value of $S$ is represented by the length and colour of the arrows in the velocity vector map in Figure 9.

The velocity vector maps of the six channels with different geometries and flow conditions were compared in the middle cross-section. Due to the superhydrophobic surface, it is obvious that the slip velocity was particularly higher than that in an ordinary channel. For a superhydrophobic channel, the Dean vortex and its intensity were enhanced, with an

increase in the apparent contact angle. Moreover, the position of the Dean vortex was translated upward and downward as the slip length increased. The velocity on the cross-section, especially close to the upside and downside walls, also increased due to the addition of the slip boundary. The distribution of velocity in the vortex became more asymmetric compared to the no-slip wall. With the development of the Re, this enhancement and transition effect in the superhydrophobic channel became more significant.

This result indicates that the slippery effect can not only reduce the flow resistance in the microchannel but can also enhance the vortices and secondary flows. This may significantly increase the local heat transfer, especially as a consequence of local accelerations. For further application, superhydrophobic surfaces can be fabricated in heat transfer structures, such as longitudinal vortex fins and wall cavities [8,41].

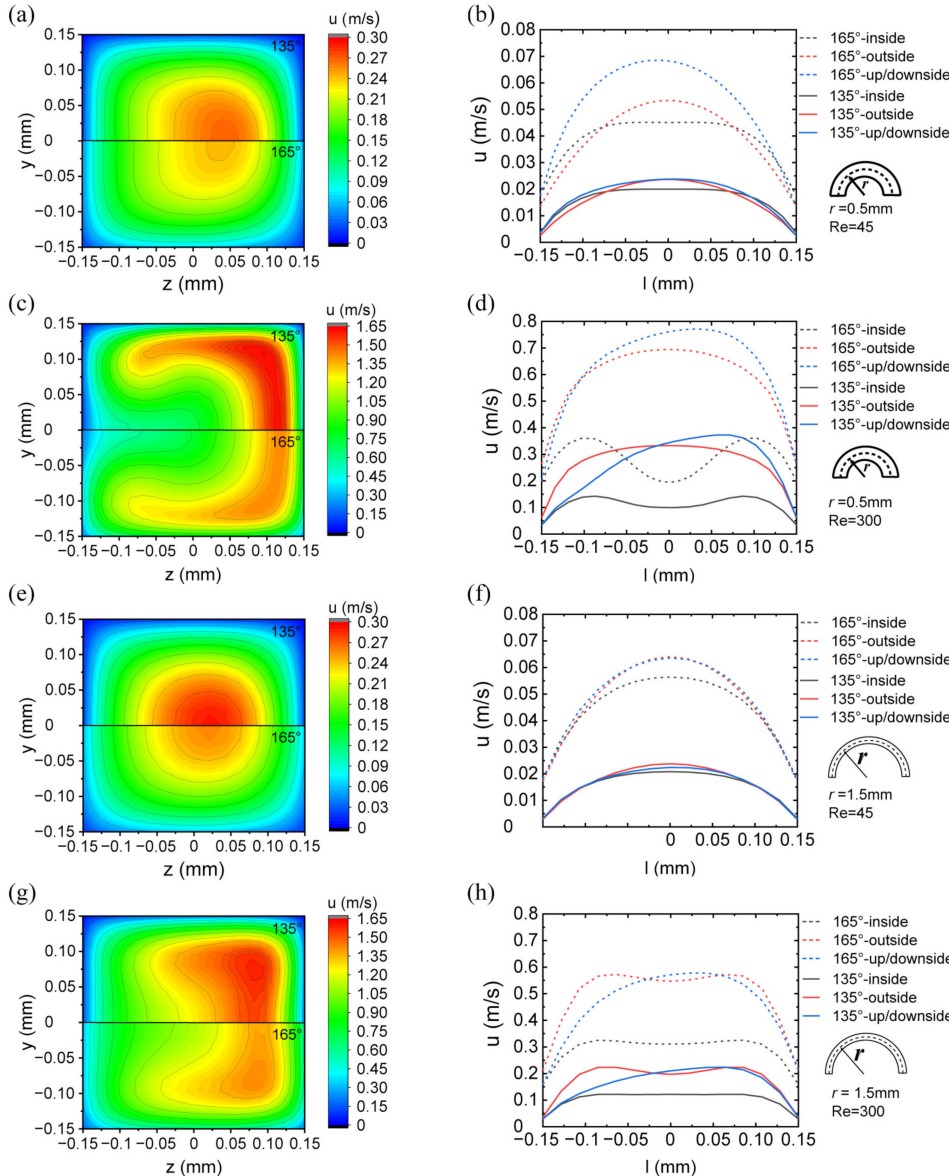

**Figure 8.** Three-dimensional non-uniform slip velocity distribution: velocity magnitude distribution color map for *r* = 0.5 mm (**a**,**c**) and 1.5 mm (**e**,**g**) and Re = 45 (**a**,**e**) and 300 (**c**,**g**), and distribution of slip velocity around the middle section of the channel for *r* = 0.5 mm (**b**,**d**) and 1.5 mm (**f**,**h**) and Re = 45 (**b**,**f**) and 300 (**d**,**h**).

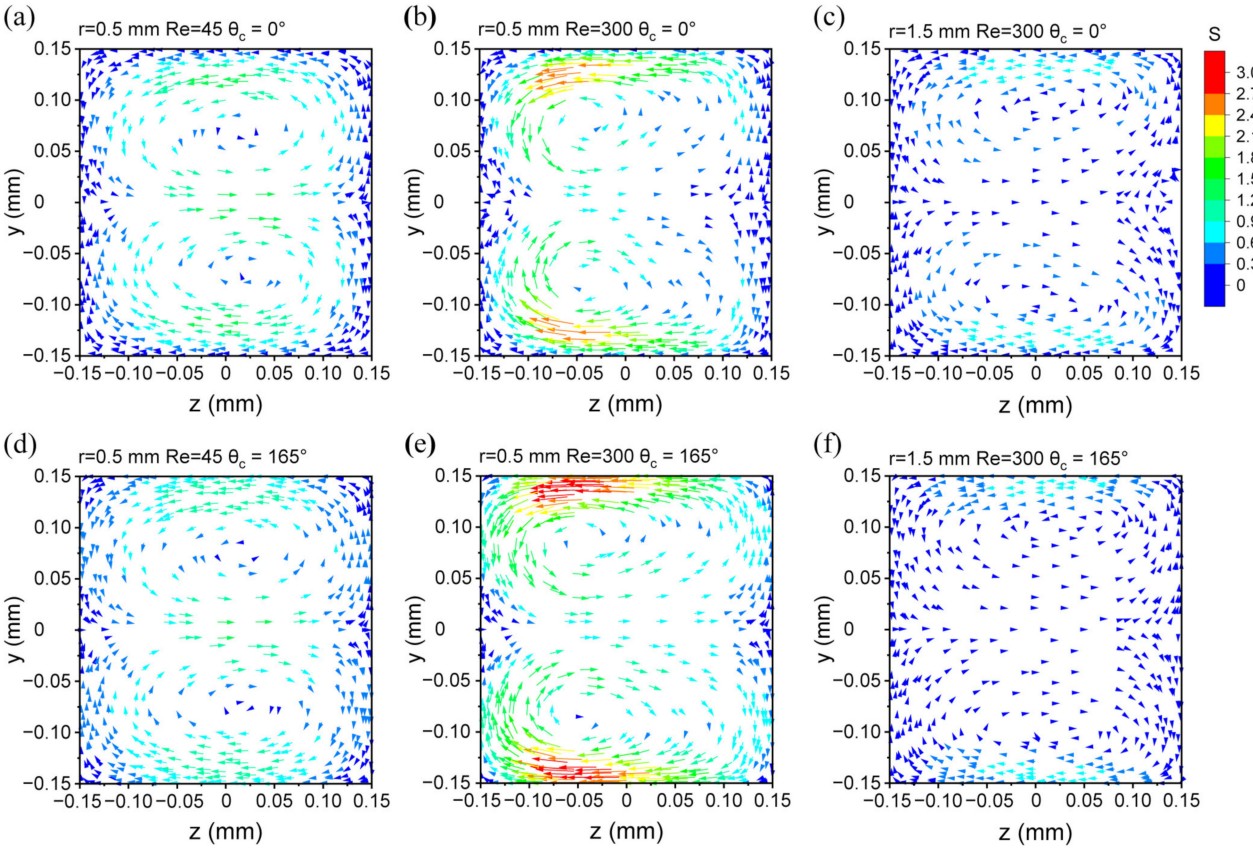

**Figure 9.** Secondary flow in middle cross-section: (**a**) $r$ = 0.5 mm, Re = 45, $\theta_c$ = 0, (**b**) $r$ = 0.5 mm, Re = 300, $\theta_c$ = 0, (**c**) $r$ = 1.5 mm, Re = 300, $\theta_c$ = 0, (**d**) $r$ = 0.5 mm, Re = 45, $\theta_c$ = 165°, (**e**) $r$ = 0.5 mm, Re = 300, $\theta_c$ = 165°, and (**f**) $r$ = 1.5 mm, Re = 300, $\theta_c$ = 165°.

### 5.3. Drag Reduction

To illustrate the differences in fluid flows of superhydrophobic channels, a comparison of *Po* and the drag reduction rate, *p*, as a function of Re and apparent contact angle is shown in Figure 10. We found that *Po* remained essentially constant when the flow rate was low, especially for a large radius of curvature. This is a reasonable result because the velocity field was almost the same as that of a straight channel, and the effect of the secondary flow can be ignored. With an increase in Re, the centrifugal effect of the flow became more significant, and *Po* rapidly grew. In this case, the maximum flow velocity region also shifted outward, which led to an increase in the slip velocity on the outside and a decrease on the inside of the microchannel. In fact, the overall drag reduction is an outcome of the balance between positive and negative effects. A superhydrophobic U-shaped channel still has a significant drag reduction, which becomes more obvious with an increase in the apparent contact angle.

As we investigated in detail, the pressure drops of a no-slip microchannel were always higher than those of a superhydrophobic microchannel. Despite a growth in the level of vorticity, the drag reduction rate gradually increased with the development of Re. In general, the drag reduction rate was in the range of 10% to 15% at $\theta_c = 135°$ and even reached 30% to 40% at $\theta_c = 165°$, which may contribute to the result of increased resistance in serpentine microchannels [42]. With an increase in the apparent contact angle, the superhydrophobic U-shaped microchannel showed better drag reduction performance, indicating huge potential for its application in complex microchannels. The variation of curvature did not play a decisive role in the overall drag reduction performance when the slip flow remained in a steady state.

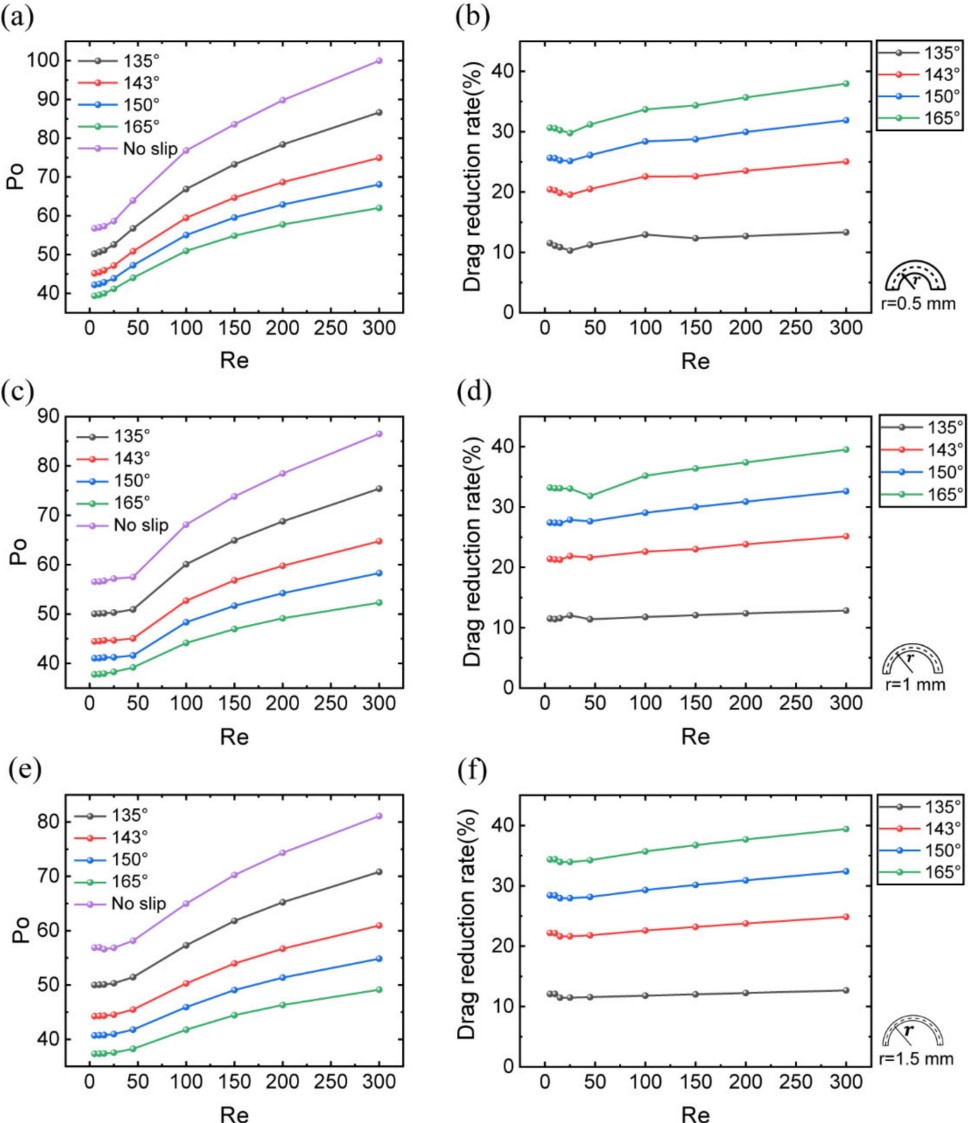

**Figure 10.** *Po* and drag reduction rate in the U-shaped microchannel: (**a,c,e**) *Po* as a function of Re for *r* = 0.5, 1, and 1.5 mm. (**b,d,f**) Drag reduction rate as a function of Re for *r* = 0.5, 1, and 1.5 mm.

## 6. Conclusions

In this study, the flow characteristics of superhydrophobic U-shaped microchannels were investigated and compared with a classical U-shaped channel, which was considered as a reference. To test the detailed fluid flow behavior, this research was extended to the typical range of Reynolds numbers from 0 to 300 with different radii of curvature. The influence of the apparent contact angle was also analyzed in detail.

The results showed that slip velocity had an obvious three-dimensional inhomogeneity on the cross-section in the U-shaped channel. This imbalance was improved with an increase in the apparent contact angle, in which the radius of curvature and flow rate were also considered. The existence of a slippery boundary enhanced the intensity of the secondary flow, which became more obvious with the increasing Re and decreasing radius of curvature. The overall drag reduction was over 30%, especially when the apparent contact angle exceeded 165°. A consistent and excellent drag reduction was achieved under different flow conditions in the superhydrophobic U-shaped microchannel. The difference of curvature did not have a distinguishing impact on the performance when the hydrophobic surface remained in a steady state. Comparing with the experiment, the numerical model assumed that the slip length remained constant in all flow conditions,



which represents that the hydrophobic surface remained in the Cassie state. If the external disturbance cannot be neglected, the overall performance may be weakened in contrast to the simulation results. This study can promote the application of superhydrophobic properties in micro-heat exchangers and the fusion process of pharmaceutical and biomedical analysis.

**Author Contributions:** Conceptualization, Z.T. and W.F.; methodology, H.L.; software, H.L.; validation, W.F., Y.H. and H.W.; formal analysis, Z.T.; investigation, W.F.; resources, T.X.; data curation, M.L.; writing—original draft preparation, Z.T.; writing—review and editing, T.X.; visualization, W.F.; supervision, T.X.; project administration, T.X.; funding acquisition, T.X. All authors have read and agreed to the published version of the manuscript.

**Funding:** This research was funded by the Natural Science Foundation of Beijing Municipality (Funder: H.L., No. JQ20016), the Beijing Science and Technology Planning Project (Funder: H.L., No. Z201100007720006), and the National Natural Science Foundation of China (Funder: T.X., No. 52006003).

**Institutional Review Board Statement:** Not applicable.

**Informed Consent Statement:** Not applicable.

**Data Availability Statement:** The data supporting the findings of this study are available from the corresponding author, Tiantong Xu, upon reasonable request.

**Conflicts of Interest:** The authors declare no conflict of interest.

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
