# Peer review of "Investigating the Flow Characteristics of Superhydrophobic U-Shaped Microchannels"

_machines, doi:10.3390/machines11010051_

Round 1

Reviewer 1 Report

Review attached.

Reviewer 2 Report

The authors report the results of numerical analyses on the effects of superhydrophobic walls on flow in microchannels, in the presence of curves. The non-uniformity of the slip condition at the walls leads to a change in the secondary flows in the curve. The topic is of interest but the presentation lacks clarity. In particular, it is only after a careful reading that one realises that the experiments to which the authors refer are only numerical, or have been carried out and used by others in the formulation of the problem. From this point of view, an exclusively numerical analysis is inconclusive in the sense that it does not provide a certain answer to the problem but only indicates the consequences of possible interpretative models of the wall slip.

In this respect, the paper is more of a technical report or an advanced exercise than a research paper. In any case, even the presentation of the numerical results lacks sensitivity analysis: how do the results change if the geometry of the microchannels does not exactly match that of the numerical model? What are the effects of slip length variability? What are the effects of corners in the cross-section of the microchannel in modulating the slip length? Ultimately, everything represented in the diagrams would appear to be certain, whereas, in reality, it is affected by variability due to a thousand causes and which has not been interpreted and described by the authors.
From this point of view, a thorough revision is requested, with an analysis on the basis of what is reported, for example, in Coleman, H. W., & Steele, W. G. (2018). Experimentation, validation, and uncertainty analysis for engineers. John Wiley & Sons., or retrievable in literature publications such as, for example, in DOI: 10.1016/j.jnnfm.2013.07.008, where experimental results are extensively used for uncertainty inference.

All data must be presented with confidence limits and uncertainty bars.

Minor comments

Figure 9 is not clear, it is suggested to reduce the number of arrows and to increase their length.

Units of measurements in the figures must be written in normal font, not in italic

Eq. 10 is dimensionally wrong.

Reviewer 3 Report

Machines

Manuscript ID : machines-2074361

Title:  Investigating the flow characteristics of superhydrophobic U-shaped microchannels

Reviewer report

This work numerically explores the the flow characteristics of a superhydrophobic U-shaped microchannel are investigated.

The topic of this paper is important. However, it needs a major revision. The following recommendations must be addressed satisfactory:

1-      Authors should show any special novelty in the work.

2-      More aspects relating the practical applicability of the work are recommended to be included in the manuscript.

3-      The mathematical formulation of the physical problem must be detailed in a clear manner.

4-      More details of the numerical method should be clearly presented.

5-      The detail of grid test must be added.

6-      The validation of the result with the previous works should be showed clearly.

Round 2

Reviewer 2 Report

The manuscript can be accepted in the revised version.

Author Response

Thank you for your decision and constructive comments on our manuscript " Investigating the flow characteristics of superhydrophobic U-shaped microchannels" (machines-2074361).  We have carefully considered the suggestion of Reviewer and make some changes. We have tried our best to improve and made some changes in the manuscript.

We sincerely hope that this revised manuscript has addressed all your comments and suggestions. We appreciated for reviewers' warm work earnestly, and hope that the correction will meet with approval.

We would like to thank the referee again for taking the time to review our manuscript.